# BRIDGING THE FAIRNESS DIVIDE: ACHIEVING GROUP AND INDIVIDUAL FAIRNESS IN GRAPH NEURAL NETWORKS

## ABSTRACT

Graph neural networks (GNNs) have emerged as a powerful tool for analyzing and learning from complex data structured as graphs, demonstrating remarkable effectiveness in various applications, such as social network analysis, recommendation systems, and drug discovery. However, despite their impressive performance, the fairness problem has increasingly gained attention as a crucial aspect to consider. Existing research on fairness in graph learning primarily emphasizes either group fairness or individual fairness; however, to the best of our knowledge, none of these studies comprehensively address both individual and group fairness simultaneously. In this paper, we propose a new concept of individual fairness within groups and a novel framework named Fairness for Group and Individual (FairGI), which considers both group fairness and individual fairness within groups in the context of graph learning. FairGI employs the similarity matrix of individuals to achieve individual fairness within groups, while leveraging adversarial learning to address group fairness in terms of both Equal Opportunity and Statistical Parity. The experimental results demonstrate that our approach not only outperforms other state-of-the-art models in terms of group fairness and individual fairness within groups, but also exhibits excellent performance in population-level individual fairness, while maintaining comparable prediction accuracy.

## 1 INTRODUCTION

Graph-based data provides a natural way to present complex relationships and structures in the real world and has wide applications in various domains, such as social networks and recommendations. Graph Neural Networks (GNNs) have emerged as powerful tools for graph-structured data, including Graph Convolutional Networks (GCNs) (Kipf & Welling, 2016), Graph Attention Neural Networks (GAT) (Wang et al., 2019), and Graphsage (Hamilton et al., 2017). Despite the impressive performance of these models, a notable limitation is that GNNs can potentially be biased and exhibit unfair prediction when the training graph is biased or contains sensitive information.

Existing work on fair graph learning mainly focuses on group, individual, and counterfactual fairness. Models emphasizing group fairness concentrate on mitigating bias at the demographic group level and guaranteeing fairness for protected groups such as FairGNN (Dai & Wang, 2021). Models prioritizing individual fairness aim to ensure fairness at the individual level, such as InFoRM (Kang et al., 2020). Graph learning with counterfactual fairness assesses fairness by assuring the fairness of predictions for each individual compared to counterfactual scenarios, like GEAR (Ma et al., 2022).

When focusing on a single type of fairness, like individual or group fairness, the existing fair graph learning models demonstrate effectiveness in mitigating bias while achieving comparable accuracy. However, group and individual fairness possess inherent limitations, and integrating them is not a trivial task. Group fairness measurements such as Statistical Parity (SP) (Dwork et al., 2012) and Equal Opportunity (EO) (Hardt et al., 2016) only consider fairness at the demographic level, neglecting individual-level fairness. For instance, Fig. 1 illustrates an admission model example where gender is the sensitive attribute, with red representing females and blue denoting males. The solid icon denotes students who meet the qualifications for admission, representing the true label, while the unfilled icon signifies those who are unqualified. In this scenario, the machine learning model has

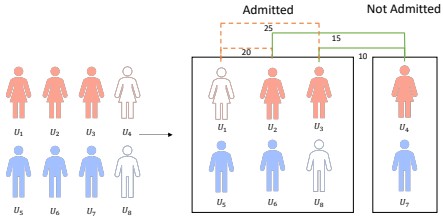

Figure 1: A toy example for student admission model with gender as the sensitive attribute. The red color represents female students, which is also the protected group. The blue color denotes male students. Solid icons correspond to students who are qualified for admission, serving as the true label, and unfilled icons denote the unqualified students.

predicted six students to be admitted and two not to be admitted. The model in the example ensures group fairness, as it can achieve the minimum EO and SP. However, the model can not guarantee individual fairness. It's because, within the female student group, student $U_4$ has a smaller distance to students $U_2$ and $U_3$ compared to student $U_1$. Thus, from the perspective of individual fairness, student $U_4$ should be admitted rather than student $U_1$. On the other hand, individual fairness utilizes the Lipchitz condition to ensure similar individuals have similar outcomes, which may lead to bias on different demographic groups (Dwork et al., 2012). GUIDE (Song et al., 2022) equalizes the individual fairness level among different groups but does not consider group fairness. In this work, we aim to mitigate unfairness at both group and individual levels and address the issue in Fig. 1.

To address the abovementioned problems, we design an innovative definition to quantify individual fairness within groups. Further, we develop a novel framework named Fairness for Group and Individual (FairGI), designed to address concerns related to group and individual fairness. Our goal is to address two major challenges: (1) how to resolve the conflicts between group fairness and individual fairness and (2) how to ensure both group fairness and individual fairness within groups in graph learning. For the first challenge, we define a new definition of individual fairness within groups. This has been designed to prevent discrepancies between group fairness and individual fairness across different groups. For the second challenge, we develop a framework FairGI to simultaneously achieve group fairness and individual fairness within groups and maintain comparable accuracy for the model prediction.

The primary contributions of this paper can be summarized as follows: (1) We introduce a novel problem concerning the achievement of both group fairness and individual fairness within groups in graph learning. To the best of our understanding, this is the first study of this unique issue; (2) We propose a new metric to measure individual fairness within groups for graphs; (3) We propose an innovative framework FairGI, to ensure group fairness and individual fairness within groups in graph learning and maintaining comparable model prediction performance; (4) Comprehensive experiments on various real-world datasets demonstrate the effectiveness of our framework in eliminating both group and individual fairness and maintaining comparable prediction performance. Moreover, the experiments show that even though we only constrain individual fairness within groups, our model achieves the best population individual fairness compared to state-of-the-art models.

## 2 RELATED WORK

### 2.1 FAIRNESS IN MACHINE LEARNING

With the advances in machine learning, the applications of machine learning models are widely used in our daily life, including financial services Leo et al. (2019), hiring decisions Chalfin et al. (2016), precision medicine MacEachern & Forkert (2021), and so on. Machine learning models can also be applied in sensitive situations and make crucial decisions for different people. However, recent research shows that the current machine learning models may suffer from discrimination Dressel & Farid (2018). Thus, considering fairness in machine learning models is becoming an important topic when we apply the models to make decisions in our daily life.

The algorithms for fairness in machine learning can be divided into three groups: pre-processing, in-processing, and after-processing. The pre-processing methods mainly focus on adjusting the input data to reduce the unfairness before training the model. The data preprocessing techniques in-

clude: re-weighting the sensitive groups to avoid discrimination Kamiran & Calders (2012) and re-sampling the data distribution Calmon et al. (2017). The in-processing methods integrate fairness constraints directly into the learning algorithm during the training process, such as adversarial debiasing Zhang et al. (2018) and fairness-aware classifier Zafar et al. (2017). Post-processing techniques adjust the output of the machine learning model after training to satisfy fairness constraints, including rejecting option-based classification Kamiran et al. (2012) and equalized odds post-processing Hardt et al. (2016).

## 2.2 Fairness in Graph Learning

GNNs are successfully applied in many areas. However, they have fairness issues because of built-in biases like homophily, uneven distributions, unbalanced labels, and old biases like gender bias. These biases can make disparities worse through unfair predictions. So, making GNNs fair is crucial for making unbiased decisions and equal outcomes. Many methods have been suggested to address fairness in learning from graphs, mainly focusing on three types: group fairness, individual fairness, and counterfactual fairness.

Group fairness methods, like FairGNN (Dai & Wang, 2021) and FairAC (Guo et al., 2023), work to make things fair at the group level. Individual fairness methods, like InForm (Kang et al., 2020) and PFR (Lahoti et al., 2019), aim to make things fair for each person. They make sure similar data points in the graph are treated the same way, no matter their group. Guide (Song et al., 2022) uses group information, but it mainly focuses on individual fairness and does not consider group fairness. Counterfactual fairness methods explore situations where some attributes or connections are changed using causal inference. This includes methods like GEAR (Ma et al., 2022) and Nifty(Agarwal et al., 2021) for graph learning.

Existing methods often overlook the simultaneous consideration of group and individual fairness, limiting their overall effectiveness. Conflicts arise between group fairness and individual fairness when the model is singularly concentrated on accomplishing both objectives (Dwork et al., 2012). Our FairGI framework remedies this by ensuring both group and individual fairness within groups while maintaining prediction accuracy. We introduce a new metric, 'individual fairness within groups,' optimize it using a node similarity matrix, and employ adversarial learning to enhance group fairness related to EO and SP.

## 3 Preliminary

### 3.1 Preliminaries for fairness learning in graphs

#### 3.1.1 Individual fairness

Individual fairness emphasizes fairness at the individual level, ensuring that individuals with similar inputs are treated consistently and fairly. We present the definition of individual fairness in graph learning based on the Lipschitz continuity below (Kang et al., 2020).

**Definition 1.** *(Individual Fairness.) Let $\mathcal{G} = (\mathcal{V}, \mathcal{E})$ be a graph with node set $\mathcal{V}$ and edge set $\mathcal{E}$. $f_G$ is the graph learning model. $Z \in R^{n \times n_z}$ is the output matrix of $f_G$, where $n_z$ is the embedding dimension for nodes, and $n = |V|$. $M \in R^{n \times n}$ is the similarity matrix of nodes. The model $f_G$ is individual fair if its output matrix $Z$ satisfies*

$$L_{If}(Z) = \frac{\sum_{v_i \in \mathcal{V}} \sum_{v_j \in \mathcal{V}} \|\mathbf{z}_i - \mathbf{z}_j\|_F^2 M[i,j]}{2} = Tr(Z^T L Z) \leq m\epsilon, \tag{1}$$

*where $L \in R^{n \times n}$ is the Laplacian matrix of $M$, $\epsilon \in R^+$ is a constant and $m$ is the number of nonzero values in $M$. $\mathbf{z}_i$ is the $i$th row of matrix $Z$ and $M[i,j]$ is the element in the $i$th row and $j$th column of matrix $M$. $L_{If}$ can be viewed as the population individual bias of model $f_G$.*

#### 3.1.2 Group fairness

In this paper, we consider two key definitions of group fairness, which are Statistical Parity (SP) (Dwork et al., 2012) and Equal Opportunity (EO) (Hardt et al., 2016).

**Definition 2.** *(Statistical Parity.)*

$$\Delta SP = P(\hat{y} = 1 | s = 0) - P(\hat{y} = 1 | s = 1), \tag{2}$$

*where $\hat{y}$ is the predicted label, $y$ is the ground truth of the label, and $s$ is the sensitive attribute.*

**Definition 3.** *(Equal Opportunity.)*

$$\Delta EO = P(\hat{y} = 1 | y = 1, s = 0) - P(\hat{y} = 1 | y = 1, s = 1), \tag{3}$$

*where $\hat{y}$ is the predicted label, $y$ is the ground truth of the label, and $s$ is the sensitive attribute.*

## 3.2 PROBLEM FORMULATION AND NOTATIONS

### 3.2.1 NOTATIONS

Let $\mathcal{G} = (\mathcal{V}, \mathcal{E}, \mathcal{X})$ be an undirected graph, where $\mathcal{V}$ is the set of nodes and $\mathcal{E}$ is the set of edges. We have $|\mathcal{V}| = n$. Let $X \in R^{n \times d}$ be the input matrix of node features with $d$ dimensions. In this paper, we assume the dataset contains a single sensitive feature characterized by binary values. Let $s_i$ be the sensitive attribute of the $i$th node in $\mathcal{G}$ and $\mathcal{S} = \{s_1, s_2, ..., s_n\}$. Let $y_i$ be the target label of the $i$th node in $\mathcal{G}$. The sensitive attribute divides the nodes into two groups. Without loss of generation, we name the group with $s = 1, s \in \mathcal{S}$ as protected group and $s = 0, s \in \mathcal{S}$ as unprotected group. Still, our methods can be easily extended to sensitive attributes with multiple values.

In this work, we address the unfairness issues in the node classification task on graphs. Our method ensures both group fairness and individual fairness within groups while preserving comparable accuracy performance. The fair graph learning problem is defined below.

### 3.2.2 PROBLEM

*Let $\mathcal{G} = (\mathcal{V}, \mathcal{E}, \mathcal{X})$ be an undirected graph with sensitive attribute $s \in \mathcal{S}$. Denote $X \in R^{n \times d}$ as the matrix of node features. Let $\mathcal{P}$ be the set of groups of nodes in $\mathcal{G}$ divided by $\mathcal{S}$, i.e. $\mathcal{V} = \mathcal{V}_{p_2} \bigcup \mathcal{V}_{p_2}$, where $\mathcal{V}_{p_i} = \{v_i | v_i \in \mathcal{V}, s_i = 0, s_i \in \mathcal{S}\}$ and $\mathcal{V}_{p_2} = \{v_i | v_i \in V, s_i = 1, s_i \in \mathcal{S}\}$. A function $f_G : G \to R^{n \times d_h}$ learns the node embeddings of $G$, i.e.,*

$$f_G(\mathcal{G}, \mathcal{S}) = H, \tag{4}$$

*where $H \in R^{n \times d_h}$ is the node embedding matrix, $|V| = n$ and $d_h$ is the dimension of node embeddings. $f$ satisfies group fairness and individual fairness within groups if and only if $H$ does not contain sensitive information and for $\forall p_k \in \mathcal{P}$,*

$$\|\mathbf{h}_i - \mathbf{h}_j\|_2 \leq c_k \|\mathbf{x}_i - \mathbf{x}_j\|_2, \ \forall i, j \in \mathcal{V}_{p_k}, \tag{5}$$

*where $\mathbf{h}_i$ is $i$th node embedding learned from $f$, $\mathbf{x}_i$ is the $i$th node feature from $X$ and $c_k \in R^+$ is the Lipschitz constant for group $p_k$.*

## 4 METHODOLOGY

### 4.1 FRAMEWORK

We propose a novel framework that balances group fairness and individual fairness in the groups to address the problem shown in figure 1 and provide a more precise measurement for group fairness. Both group fairness and individual fairness have limitations in real-world application. While group fairness focuses on fairness at the group level, it ignores individual fairness within these groups. Current individual fairness approaches measure individual fairness by Lipchitz Continous (Kang et al., 2020). However, strictly adhering to individual fairness may lead to conflicts with group fairness (Dwork et al., 2012). Our method provides a novel framework to address the above challenges by proposing a novel definition of individual fairness within groups. Combining this new definition with group fairness, we develop a more precise and reliable approach to guarantee fairness in graph learning.

The detailed algorithm of FairGI is shown in Algorithm 1. Since our method allows for model training even when sensitive labels are partly missing, we initially train a sensitive attribute estimator

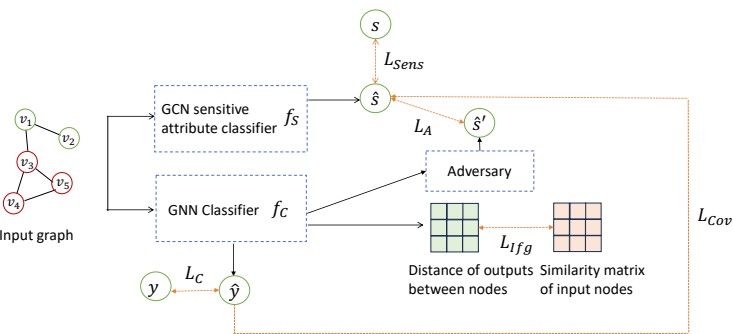

Figure 2: Overview of FairGI. Our method comprises four parts, i.e., a GNN classifier for node prediction, a sensitive attribute classifier for sensitive attribute prediction, an adversary layer, and an individual fairness module.

---

**Algorithm 1** Algorithm of our framework

---

1: **Input:** $\mathcal{G}(\mathcal{V}, \mathcal{E}), X, \mathcal{S}$
2: **Output: Sensitive attribute classifier** $f_S$**, node classifier** $f_C$**, node label prediction** $\hat{y}$
3: Train sensitive attribute classifier $f_S$ by given sensitive attribute labels using loss function $L_{Sens}$ in Eq.(11).
4: **repeat**
5:     Obtain estimated sensitive attribute $\hat{s}, \hat{s} \in \tilde{\mathcal{S}}$ by $f_S$.
6:     Optimize $f_G$ to predict the node label by loss function $L_C$ in Eq.(19)
7:     Optimize $f_G$ to debias group unfairness by loss function $L_G$ in Eq.(18)
8:     Optimize $f_G$ to debias individual unfairness within groups by loss function $L_{Ifg}$ in Eq.(10)
9:     Optimize adversary $f_A$ by $L_A$ in Eq.(23)
10: **until** Converge
11: **return** $f_S$, $f_G$ and $\hat{y}$

---

utilized GCN (Kipf & Welling, 2016) to predict the unlabeled sensitive attributes. For the node classification task, we employ GAT (Wang et al., 2019) to generalize node embedding and predict target labels. We design a novel loss function to achieve individual fairness within groups in our framework. To guarantee group fairness, we employ an adversarial learning layer that hinders adversaries from precisely predicting sensitive attributes, thereby reducing the bias from sensitive information. Unlike FairGNN (Dai & Wang, 2021), which solely optimizes SP, we theoretically demonstrate that our adversarial loss function can enhance group fairness in terms of both EO and SP. In addition, we devise a conditional covariance constraint loss function to increase the stability of the adversarial learning process and prove that optimizing the loss function leads to the minimum value of EO.

Figure 2 shows the framework of FairGI. Our method initially trains a sensitive attribute classifier $f_S$ and predicts the unlabelled sensitive attributes. To ensure individual fairness within groups, we introduce a loss function designed to minimize bias between individuals in the same group. Additionally, to address group unfairness, we employ adversarial learning and covariance loss functions to reduce EO and SP.

The comprehensive loss function of our approach is:

$$L = L_C + L_G + \alpha L_{Ifg}, \tag{6}$$

where $L_C$ is the loss for node label prediction, $L_G$ represents the loss for debiasing group unfairness, and $L_{Ifg}$ represents the loss for mitigating individual unfairness within groups.

### 4.2 OPTIMIZATION OF INDIVIDUAL FAIRNESS WITHIN GROUPS

#### 4.2.1 CHALLENGES OF BALANCING INDIVIDUAL FAIRNESS AND GROUP FAIRNESS

We can observe that the definitions of group fairness, especially SP and EO, may contradict individual fairness in specific circumstances. As noted by (Dwork et al., 2012), potential discrepancies can arise between group and individual fairness when the distance between groups is significant.

Assume $\Gamma$ represents the protected group and $\Gamma'$ denotes the unprotected group. If there is a considerable distance between individuals in $\Gamma$ and those in $\Gamma'$, strictly adhering to individual fairness may not guarantee similar outcomes for both groups. Thus, this can lead to a conflict with the goal of group fairness, which seeks to maintain equal treatment and opportunity for all groups. Inspired by (Dwork et al., 2012), we alleviate the conflicts between SP and individual fairness by loosening the Lipschitz restriction between different groups.

### 4.2.2 Loss function for individual fairness within groups

To balance group and individual fairness and address the limitation of group fairness, we proposed a loss function that ensures individual fairness within the known groups. We first propose the definition of individual fairness within groups. Note that our approach is also applicable to groups that are not mutually exclusive.

**Definition 4.** *(Individual Fairness within Groups.) The measurement of individual fairness within group $p$ is:*

$$L_p(Z) = \frac{\sum_{v_i \in \mathcal{V}_p} \sum_{v_j \in \mathcal{V}_p} \|\mathbf{z}_i - \mathbf{z}_j\|_2^2 M[i,j]}{n_p}, \tag{7}$$

*where $\mathcal{V}_p$ represents the set of nodes in group $p$ and $Z$ denotes the node embedding matrix. $\mathbf{z}_i$ is the $i$th row of $Z$. $M$ is the similarity matrix of nodes. $L$ is the laplacian matrix of $M$ and $n_p$ is the number of pairwise nodes in group $p$ with nonzero similarities in $M$.*

Since we aim to contain individual fairness in each group to ensure the treatment of individuals within the groups is fair, we consider minimizing the maximum individual unfairness over all groups in our loss function.

Firstly, we consider the loss function that minimizes the maximum unfairness over all the groups as follows:

$$f^* = \text{argmin}_{f \in \mathcal{H}} \{ max_{p \in \mathcal{P}} L_p(Z) \}, \tag{8}$$

where $Z = f(\cdot)$, $\mathcal{H}$ is a class of graph learning models, $\mathcal{P}$ is the set of groups and $L_p$ is the individual loss for group $p$. Motivated by the concept of guaranteeing the optimal situation for the most disadvantaged group, as presented in (Diana et al., 2021), we employ the minimax loss function from Eq. (8). This approach prioritizes minimizing the maximum unfairness across groups, rather than simply aggregating individual fairness within each group to ensure a more equitable outcome.

The optimal solution in Eq.(8) is hard to obtain, thus we can relax the loss function as expressed in Eq. (9). Given an error bound $\gamma$ for each group, the extension of the minimax problem can be formulated as follows:

$$\text{minmize}_{f \in \mathcal{H}} \sum_{p \in \mathcal{P}} L_p(Z), \quad \text{subject to } L_p(f) \leq \gamma, p \in \mathcal{P}. \tag{9}$$

In our framework, we not only focus on the loss of individual fairness within groups in Eq.(9), but also consider the loss of label prediction and group fairness. Thus, we further convert Eq.(9) into the unconstrained loss function shown in Eq.(10) by introducing Lagrange multiplier $\lambda_p$ to the loss function. We can achieve individual fairness within groups by minimizing the loss function below:

$$L_{Ifg} = \sum_{p \in \mathcal{P}} L_p(Z) + \sum_{p \in \mathcal{P}} \lambda_p (L_p(Z) - \gamma), \tag{10}$$

where $\lambda_p$ and $\gamma$ are hyperparameters in our model.

### 4.3 Enhancing Group Fairness Through Ensuring Equal Opportunity and Statistical Parity

In this section, we improve group fairness by considering both EO and SP. Different from FairGNN (Dai & Wang, 2021), which only emphasizes optimizing SP, our method is designed to optimize both EO and SP simultaneously.

### 4.3.1 ADVERSERIAL LEARNING

As we address the circumstance where certain sensitive labels are absent, we utilized GCN (Kipf & Welling, 2016) to train the sensitive estimator $f_S$, and the loss function for the sensitive label prediction is:

$$L_{Sens} = -\frac{1}{|\mathcal{V}|} \sum_{i \in \mathcal{V}} ((s_i)\log(\hat{s}_i) + (1 - s_i)\log(1 - \hat{s}_i)), \quad (11)$$

where $s_i$ is the sensitive attribute for the $i$th node, $\hat{s}_i$ is the predicted senstive labels.

To optimize SP, the min-max loss function of adversarial learning is (Dai & Wang, 2021):

$$\min_{\Theta_C} \max_{\Theta_A} L_{A_1} = \mathbb{E}_{h \sim p(h|\hat{s}=1)}[\log(f_A(h))] + \mathbb{E}_{h \sim p(h|\hat{s}=0)}[\log(1 - f_A(h))], \quad (12)$$

where $\Theta_C$ is the parameters for graph classifier $f_C$, $\Theta_A$ is the parameters for adversary $f_A$ and $h$ is the node presentation of the last layer of GNN classifier $f_C$. $h \in p(h|\hat{s} = 1)$ denotes sampling nodes from the protected group within the graph $\mathcal{G}$.

FairGNN demonstrates that optimizing Eq. (12) can achieve the minimum SP (Dai & Wang, 2021) in the GNN classifier. However, it does not guarantee the attainment of the minimum EO. While both EO and SP are significant metrics for group fairness, optimizing solely for SP can adversely affect the performance of EO, leading to model bias. We propose a novel min-max loss function designed for adversarial learning to achieve the minimum EO in Eq. (13).

$$\min_{\Theta_C} \max_{\Theta_A} L_{A_2} = \mathbb{E}_{h \sim p(h|\hat{s}=1,y=1)}[\log(f_A(h))] + \mathbb{E}_{h \sim p(h|\hat{s}=0,y=1)}[\log(1 - f_A(h))]. \quad (13)$$

The Theorem 6 in the Appendix demonstrates that the optimal solution of Eq. (13) ensures the GNN classifier satisfies $\Delta EO = 0$, given two easily attainable assumptions. In addition, we can also mitigate sensitive information by letting $f_A$ predict the sensitive attribute closer to a uniform distribution, as inspired by Gong et al. (2020).

Combining Eq.(12) and Eq.(13), we have the loss function of adversarial learning as:

$$L_A = L_{A_1} + L_{A_2}. \quad (14)$$

### 4.3.2 COVARIANCE CONSTRAINT

The limitation of adversarial debiasing is instability. Similar to adversarial learning, FairGNN only considers optimizing SP in the covariance constraint loss as below (Dai & Wang, 2021).

$$L_{R_1} = |Cov(\hat{s}, \hat{y})| = |\mathbb{E}[(\hat{s} - \mathbb{E}[\hat{s}])(\hat{y} - \mathbb{E}[\hat{y}])]|, \quad (15)$$

However, Eq. (15) dose not consider EO, and $\Delta EO = 0$ is not the prerequisite of $L_{R_1} = 0$, which damages the model performance in EO.

Thus, we propose a covariance constraint loss function for optimizing EO as follows:

$$L_{R_2} = |Cov(\hat{s}, \hat{y}|y = 1)| = |\mathbb{E}[(\hat{s} - \mathbb{E}[\hat{s}|y = 1](\hat{y} - \mathbb{E}[\hat{y})|y = 1]|y = 1]|. \quad (16)$$

Theorem 7 in the Appendix shows that under the mild assumption, $L_{R2} = 0$ is the prerequisite of $\Delta EO = 0$. We can enhance group fairness in our model by optimizing EO and SP using Eq. (17).

$$L_{Cov} = L_{R1} + L_{R2}. \quad (17)$$

In conclusion, the loss function that we utilize to mitigate group fairness is:

$$L_G = \beta L_A + \gamma L_{Cov}, \quad (18)$$

where $\beta$ and $\gamma$ are hyperparameters.

## 4.4 NODE PREDICTION

For the node prediction task, we employed GAT (Wang et al., 2019) to predict node labels. The loss function for GNN classifier $f_C$ is:

$$L_C = -\frac{1}{|\mathcal{V}|} \sum_{i \in \mathcal{V}} ((y_i)\log(\hat{y}_i) + (1 - y_i)\log(1 - \hat{y}_i)). \quad (19)$$

Table 1: Comparisons of our method and baselines on three datasets. ↑ denotes the larger is the better and ↓ indicates the smaller is the better. Best performances are in bold.

| Dataset | Method | Acc ↑ | AUC ↑ | $\Delta SP$ ↓ | $\Delta EO$ ↓ | MaxIG ↓ | IF ↓ |
|---|---|---|---|---|---|---|---|
| | GCN | 68.82±0.17 | 73.98±0.07 | 2.21±0.61 | 3.17±1.10 | 5.69±0.08 | 899.54±13.10 |
| | GAT | 69.14±0.68 | 74.24±0.90 | 1.40±0.64 | 2.86±0.49 | 6.10±0.62 | 880.89±89.97 |
| | PRF | 55.39±0.08 | 53.83±0.02 | 1.08±0.09 | 1.82±0.18 | 0.64±0.01 | 101.26±1.27 |
| | InFoRM | 68.77±0.39 | 73.69±0.10 | 1.84±0.69 | 3.58±1.15 | 1.52±0.05 | 238.41±7.97 |
| Pokec-n | NIFTY | 65.97±0.57 | 69.87±0.64 | 4.62±0.52 | 7.32±0.94 | 1.87±0.15 | 310.84±26.25 |
| | GUIDE | 69.46±0.04 | 74.67±0.01 | 2.95±0.11 | 0.80±0.18 | 0.61±0.00 | 101.77±0.28 |
| | FairGNN | **69.86±0.30** | **75.58±0.52** | 0.87±0.38 | 2.00±1.08 | 1.26±0.96 | 192.29±142.06 |
| | Ours | 68.86±0.58 | 75.07±0.1 | **0.63±0.37** | **0.75±0.30** | **0.47±0.09** | **67.41±13.68** |
| | GCN | 69.08±2.02 | 74.20±1.69 | 17.12±7.10 | 10.03±4.92 | 25.99±2.12 | 17.87±1.74 |
| | GAT | 70.80±3.70 | 72.48±4.32 | 11.90±8.94 | 16.70±10.57 | 21.14±10.86 | 20.79±9.95 |
| | PRF | 55.58±0.93 | 58.26±4.45 | 1.99±0.99 | 2.22±1.65 | 4.47±2.25 | 3.06±1.55 |
| | InFoRM | 68.71±2.78 | 74.19±1.85 | 16.64±5.64 | 12.75±6.80 | 26.52±8.25 | 18.82±5.17 |
| NBA | NIFTY | 70.55±2.30 | 76.18±0.83 | 11.82±4.28 | 5.69±3.48 | 17.14±5.01 | 11.93±3.43 |
| | GUIDE | 63.31±2.86 | 67.46±3.44 | 13.89±5.11 | 10.50±4.76 | 29.54±16.34 | 19.84±10.75 |
| | FairGNN | 72.95±2.10 | 77.37±1.11 | 1.19±0.43 | **0.62±0.43** | 10.91±12.59 | 18.51±23.72 |
| | Ours | **73.13±1.75** | **79.28±0.46** | **0.43±0.28** | **0.62±0.32** | **0.12±0.11** | **0.08±0.07** |
| | GCN | 70.35±0.99 | 65.18±6.58 | 14.55±6.13 | 13.92±6.00 | 5.18±1.34 | 39.11±6.69 |
| | GAT | 70.89±1.84 | 71.30±1.64 | 15.95±2.40 | 15.96±2.77 | 5.88±3.34 | 35.28±15.41 |
| | PRF | 69.87±0.09 | 69.90±0.04 | 14.63±0.78 | 13.96±0.79 | 5.80±0.08 | 39.79±0.63 |
| | InFoRM | 69.91±3.70 | 65.55±5.6 | 14.80±3.84 | 14.82±4.18 | 4.09±1.68 | 33.62±13.86 |
| Credit | NIFTY | 68.74±2.34 | 68.84±0.41 | 9.91±0.30 | 9.07±0.63 | 2.92±1.18 | 24.73±9.75 |
| | GUIDE | 62.01±0.01 | 67.44±0.01 | 13.88±0.10 | 13.54±0.06 | **0.22±0.01** | 1.90±0.01 |
| | FairGNN | 73.40±0.15 | **70.18±0.03** | 3.91±0.11 | 3.49±0.25 | 1.88±0.09 | 13.84±0.70 |
| | Ours | **74.09±0.13** | 68.81±0.11 | **3.84±0.22** | **2.60±0.20** | **0.22±0.01** | **1.84±0.10** |

## 5 EXPERIMENTS

In this section, we conduct a comprehensive comparison between our proposed method and other cutting-edge models, evaluating their performance on real-world datasets to demonstrate the effectiveness of our approach.

### 5.1 DATASETS AND BASELINES

In this experiment, we utilize three public datasets, Pokec_n Dai & Wang (2021) , NBA Dai & Wang (2021), and Credit Yeh & Lien (2009).

We compare our method with other state-of-art fairness models for graph learning. In our comparison, we include basic GNNs like GCN (Kipf & Welling, 2016) and GAT (Velickovic et al., 2017), which don't fix bias. We also include GNNs like PFR (Lahoti et al., 2019) and InFoRM (Kang et al., 2020), aiming at individual fairness. To compare group fairness methods, we include FairGNN (Dai & Wang, 2021). Plus, we compare GNNs with causal inference fairness such as NIFTY (Agarwal et al., 2021). Further descriptions of datasets and baselines can be found in the Appendix.

### 5.1.1 EVALUATION METRICS

In this experiment, our primary focus is on analyzing and comparing individual fairness within groups as well as group fairness. Furthermore, we assess the performance of the prediction task by employing metrics such as Area Under the Curve (AUC) and Accuracy (ACC). We include MaxIG, IF, SP, and EO in the experiments for the fairness evaluation metrics. MaxIG is defined as:

$$\text{MaxIG} = \max(L_p(Z)), \; p \in \mathcal{P}, \tag{20}$$

where $L_p(\cdot)$ can be computed in Eq. (7) and $Z$ is the output of GNN Classifier.

### 5.2 RESULTS AND ANALYSIS

### 5.2.1 INDIVIDUAL UNFAIRNESS AND GROUP UNFAIRNESS IN GRAPH NEURAL NETWORKS

Based on the experimental results presented in Table 1, several key findings emerge regarding the performance and biases of various GNNs.

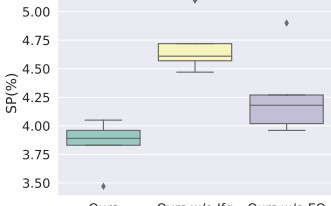 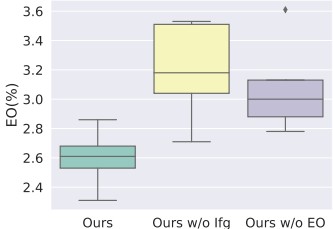

Figure 3: Comparison of our method, our method without loss function of individual fairness within groups, our method without the optimization for EO.

Traditional GNNs, such as GCN and GAT, exhibit both individual and group biases. This suggests that while these models may have good performance, they do not adequately handle fairness issues. Models that address group fairness, such as FairGNN, demonstrate good performance in group fairness metrics like SP and EO, but struggle with individual fairness metrics, such as IF and MaxIG. This underscores the challenge of simultaneously optimizing for both group and individual fairness.

On the contrary, models like PRF, InFoRM, NIFTY, and GUIDE, which primarily target individual fairness, perform well in mitigating individual biases. However, they have poor performance in group fairness. This dichotomy indicates a potential trade-off regarding group-level fairness while promoting individual fairness. These findings emphasize the need for more comprehensive solutions that simultaneously address individual and group biases.

### 5.2.2 EFFECTIVENESS OF FAIRGI IN MITIGATING BOTH INDIVIDUAL FAIRNESS AND GROUP FAIRNESS

The results in Table 1 highlight the efficacy of our approach, leading to two primary observations: (1) Our method outperforms competing methods by ensuring superior group fairness and intra-group individual fairness while retaining comparable prediction accuracy and AUC of the ROC curve; (2) While our technique is constrained only to fairness within groups, it remarkably achieves superior population individual fairness compared to baselines. This suggests that we can attain the pinnacle of population individual fairness by concentrating solely on intra-group individual fairness and overlooking inter-group individual fairness. This outcome is intuitively reasonable given the potential substantial variances among individuals from different groups.

### 5.3 ABLATION STUDIES

In the ablation study, we examine the impact of two modules, individual fairness within groups and optimization of equal opportunity, on the performance of our method. We conduct a comparison between our method and two of its variants using the Credit dataset. The first variant, "ours w/o Ifg," omits the individual fairness within groups loss $L_{Ifg}$ from our method. The second variant, "ours w/o EO," eliminates the optimizations for equal opportunity, specifically the loss functions $L_{A_2}$ and $L_{R2}$, from our method.

Figure 3 illustrates the comparative results. We can observe that upon the removal of $L_{Ifg}$, there is a noticeable increase in MaxIG, EO, and SP, with MaxIG experiencing the most significant rise. This strongly attests to the efficacy of the loss function $L_{Ifg}$ in enhancing individual fairness. When we disregard the optimizations for EO, MaxIG remains relatively unchanged while both EO and SP increase. This highlights the crucial role of $L_{A2}$ and $L_{Cov2}$ in optimizing group fairness.

### 6 CONCLUSION

In this paper, we present an innovative problem that considers both group fairness and individual fairness within groups. In this particular context, we propose a novel definition named MaxIG for individual fairness within groups. Furthermore, we propose a novel framework named FairGI to achieve both group fairness and individual fairness within groups in graph learning. FairGI leverages the similarity matrix to mitigate individual unfairness within groups. Additionally, it exploits the principles of adversarial learning to mitigate group unfairness. Extensive experiments demonstrate that FairGI achieves the best results in fairness and maintains comparable prediction performance.

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

# A  APPENDIX

## A.1  PROOF OF THEOREMS

**Proposition 5.** *Let Eq.(13) be the loss function of adversary learning. The optimal solution of Eq.(13) is achieved if and only if $p(h|\hat{s} = 0, y = 1) = p(h|\hat{s} = 1, y = 1)$.*

*Proof.* By Proposition 1. in dai2021say, the optimal value of adversary is in Eq.(21)

$$f_{A_2}^*(h) = \frac{p(h|\hat{s} = 1, y = 1)}{p(h|\hat{s} = 1, y = 1) + p(h|\hat{s} = 0, y = 1)}. \tag{21}$$

We denote $B = p(h|\hat{s} = 1, y = 1)$ and $C = p(h|\hat{s} = 0, y = 1)$. Thus, the min-max loss function in Eq.(13) can be written as the following with the optimal solution of adversary:

$$\begin{aligned} L_{A_2} &= \mathbb{E}_{h \in B}[log \frac{B}{B + C}] + \mathbb{E}_{h \in C}[log \frac{C}{B + C}] \\ &= \mathbb{E}_{h \in B}[log \frac{B}{\frac{1}{2}(B + C)}] + \mathbb{E}_{h \in C}[log \frac{C}{\frac{1}{2}(B + C)}] - 2log2 \\ &= D_{KL}(B||B + C) + D_{KL}(C||B + C) - 2log2 \\ &= 2JSD(B||C) - 2log2, \end{aligned} \tag{22}$$

where $D_{KL}(\cdot)$ is the Kullback–Leibler divergence and $JSD(\cdot)$ is the Jensen–Shannon divergence.

We know that $JSD(B||C)$ is non-negative and equals to 0 if and only if distributions $B$ and $C$ are equal. Thus, the loss function $L_{A_2}$ achieves the minimum value if and only if $p(h|\hat{s} = 0, y = 1) = p(h|\hat{s} = 1, y = 1)$. The proof is adapted to Proposition 4.1 in Dai & Wang (2021).

$\square$

**Theorem 6.** *Let $\hat{y}$ be the prediction label of GNN classifier $f_G$, $h$ be the node presentation generated by GNN classifier $f_G$. We assume:*

1. *The prediction of sensitive attribute $\hat{s}$ and $h$ are conditionally independent, i.e., $p(\hat{s}, h|s, y = 1) = p(\hat{s}|s, y = 1)p(h|s, y = 1)$.*

2. *$p(s = 1|\hat{s} = 1, y = 1) \neq p(s = 1|\hat{s} = 0, y = 1)$.*

*If Eq.(13) achieves the global optimum, the prediction of GNN classifier $f_G$ will satisfy equal opportunity, i.e. $p(\hat{y}|s = 0, y = 1) = p(\hat{y}|s = 1, y = 1)$.*

Combining Eq.(12) and Eq.(13), we have the loss function of adversarial learning as:

$$L_A = L_{A_1} + L_{A_2}. \tag{23}$$

*Proof.* By proposition 7.1, we have $p(h|\hat{s} = 0, y = 1) = p(h|\hat{s} = 1, y = 1)$ when we obtain the optimum solution for the loss function 13. Thus, we have

$$\sum_{s \in S} p(h, s|\hat{s} = 1, y = 1) = \sum_{s \in S} p(h, s|\hat{s} = 0, y = 1). \tag{24}$$

Under the conditionally independent assumption in assumption 1, we have

$$\begin{aligned} \sum_{s \in S} p(h|\hat{s} = 1, y = 1)p(s|\hat{s} = 1, y = 1) = \\ \sum_{s \in S} p(h|\hat{s} = 0, y = 1)p(s|\hat{s} = 0, y = 1). \end{aligned} \tag{25}$$

Reformulating the Eq.(25) and by the assumption 2, we obtain

$$\begin{aligned}
\frac{p(h|s=1,y=1)}{p(h=0,y=1)} &= \frac{p(s=0|\hat{s}=0,y=1) - p(s=0|\hat{s}=1,y=1)}{p(s=1|\hat{s}=1,y=1) - p(s=1|\hat{s}=0,y=1)} \\
&= \frac{1 - p(s=1|\hat{s}=0,y=1) - 1 + p(s=1|\hat{s}=1,y=1)}{p(s=1|\hat{s}=1,y=1) - p(s=1|\hat{s}=0,y=1)} \\
&= 1
\end{aligned} \tag{26}$$

Thus, we have $p(h|s=1,y=1) = p(h|s=0,y=1)$, which leads to $p(\hat{y}|s=1,y=1) = p(\hat{y}|s=0,y=1)$. The equal opportunity is satisfied when we achieve the global minimum in Eq.(13). The proof is adapted to Theorem 4.2 in Dai & Wang (2021).

$\square$

**Theorem 7.** *Suppose $p(\hat{s},h|s,y=1) = p(\hat{s}|s,y=1)p(h|s,y=1)$, when $f_G$ satisfy equal opportunity, i.e. $p(\hat{y},s|y=1) = p(\hat{y}|y=1)p(s|y=1)$, we have $L_{R_2} = 0$.*

*Proof.* Since $p(\hat{s},h|y=1) = p(\hat{s}|y=1)p(h|y=1)$, we have the following equation:

$$\begin{aligned}
p(h|s,\hat{s},y=1) &= \frac{p(h,s,\hat{s}|y=1)}{p(s,\hat{s}|y=1)} \\
&= \frac{p(\hat{s},h|s,y=1)}{p(\hat{s}|s,y=1)} \\
&= p(h|s,y=1),
\end{aligned} \tag{27}$$

thus, we have $p(\hat{y}|s,\hat{s},y=1) = p(\hat{y}|s,y=1)$.

If $p(\hat{y},s|y=1) = p(\hat{y}|y=1)p(s|y=1)$, $p(\hat{y},\hat{s}|y=1)$ can be written as:

$$\begin{aligned}
p(\hat{y},\hat{s}|y=1) &= \sum_{s \in S} p(\hat{y}|s,y=1)p(\hat{s},s|y=1) \\
&= p(\hat{y}|y=1)p(\hat{s}|y=1).
\end{aligned} \tag{28}$$

Thus, we have $L_{R_2} = |Cov(\hat{s},\hat{y}|y=1)| = 0$, which proof the theorem. The proof is adapted by the proof of Theorem 4.3 in Dai & Wang (2021). $\square$

## A.2 DATASETS AND BASELINES

**Datasets.** In the experiments, we utilize three datasets. Table 2 shows the summary of the datasets. Our datasets demonstrate comprehensive coverage of diverse data categories and varied sample sizes. The detailed descriptions of the datasets are presented below:

- The PockeC dataset, presented by Takac & Zabovsky (2012), serves as a benchmark dataset derived from Slovakian social networks, facilitating the evaluation and development of various algorithms and models in this context. This comprehensive dataset encompasses various features for each individual within the network, such as gender, age, educational background, geographical region, recreational activities, working areas, etc. (Dai & Wang, 2021) partitioned the dataset into two distinct subsets, Pokec_n and Pokec_z, based on the provinces of the individuals. Each of the two datasets contains two predominant regions within the relevant provinces. In this experiment, we utilize the Pokec_n dataset and regard the geographical region as the sensitive attribute. In the node classification task, we use working areas as the target variable for node prediction.

- The NBA dataset, introduced by Dai & Wang (2021), consists of data from 403 professional basketball players in the National Basketball Association (NBA). The dataset includes features such as age, nationality, salary, and other relevant player attributes. In our experiments, nationality is considered a sensitive attribute, while the target label focuses on determining whether a player's salary is above or below the median.

- The Credit dataset is introduced by Yeh & Lien (2009), which offers valuable insights into various aspects of consumer behavior. The dataset includes features such as spending

| Count | Pokec-n | Credit | NBA |
|---|---|---|---|
| Number of Nodes | 66,569 | 30,000 | 403 |
| Number of node attributes | 59 | 13 | 39 |
| Number of Edges | 729,129 | 304,754 | 16,570 |
| Senstive attibute | region | age | nationality |

Table 2: Basic statistics of datasets.

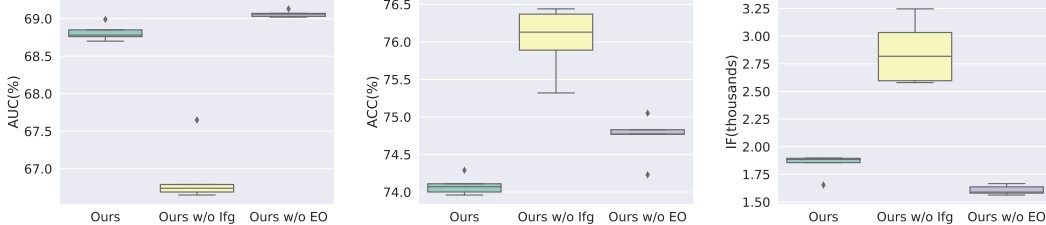

Figure 4: Comparison of our method, our method without loss function of individual fairness within groups, our method without the optimization for EO.

habits and credit history, which are essential for understanding the financial patterns of these individuals. The primary objective of this dataset is to facilitate the prediction of credit card default, with age being identified as the sensitive attribute.

**Baselines.** We compare our methods with other state-of-art models in the node classification task.

- **FairGNN**: FairGNN is a graph neural network (GNN) model introduced by Dai & Wang (2021) employs adversarial learning address the challenges of group fairness in graph representation learning.

- **GUIDE**: GUIDE was proposed by Song et al. (2022) to ensure group equality informed individual fairness in graph representation learning.

- **PRF**: Pairwise Fair Representation (PFR) is a graph learning method introduced by Lahoti et al. (2019) to achieve individual fairness in graph representation learning.

- **InFoRM**: Individual Fairness on Graph Mining (InFoRM) (Kang et al., 2020) achieves individual fairness in graph representation learning by employing Lipschitz continuity.

- **NIFTY**: Agarwal et al. (2021) propose NIFTY (unifying fairness and stability) that applies the Lipschitz condition to achieve counterfactual fairness in graph learning.

### A.3 EXPERIMENT SETTINGS

In this experiment, we compare our method to the state of art models for fairness in graph learning. Here we use Graph Convolutional Network (GCN) (Kipf & Welling, 2016) and Graph Attention Network (GAT) (Wang et al., 2019) as the vanilla comparison models since they do not apply fairness skills. We also include graph learning models with group fairness like FairGNN (Dai & Wang, 2021). Graph learning models with individual fairness include GUIDE Song et al. (2022), PRF (Lahoti et al., 2019) and InFoRM (Kang et al., 2020). Graph learning models with counterfactual fairness such as NIFTY (Agarwal et al., 2021). The parameters of our method are shown in Table 3. We apply four datasets in the experiments. For each dataset we randomly divide them into training set, test set, and validation set with ratios of 50%, 25% and 25%.

### A.4 ADDITIONAL ANALYSIS ON ABLATION STUDIES

Figure 4 presents performance comparisons based on AUC, ACC, and IF. We note that eliminating individual fairness losses within groups results in a marginal increase in ACC compared to our

| Count | Pokec-n | Credit | NBA |
|---|---|---|---|
| $\alpha$, coefficient of $L_{Ifg}$ | 1e-9 | 0.5 | 1e-9 |
| $\beta$, coefficient of $L_A$ | 0.02 | 0.8 | 0.01 |
| $\gamma$ | 0.004 | 0.004 | 0.004 |
| $\lambda_1$ | 0.5 | 0.5 | 0.5 |
| $\lambda_2$ | 1.25 | 1.25 | 1 |
| $\eta$, coefficient of $L_{Cov}$ | 3 | 6 | 16 |
| number of sensitive labels | 200 | 500 | 50 |
| learning rate | 0.0005 | 0.001 | 0.001 |
| weight decay | 1e-5 | 1e-5 | 1e-5 |

Table 3: Hyper parameter setting for datasets.

approach. When we exclude EO optimization losses, both ACC and AUC exhibit a non-significant increase, demonstrating that our method can maintain comparable accuracy.

Furthermore, We can observe that if we remove the loss of individual fairness within groups, the performance of IF becomes worse. This observation demonstrates the effectiveness of the loss function of individual fairness within groups, i.e., $L_{Ifg}$.

