# OpenReview forum: "Bridging the Fairness Divide: Achieving Group and Individual Fairness in Graph Neural Networks"
_ICLR.cc/2024/Conference — Submitted to ICLR 2024_

### Official Review · Reviewer_uySt · 2023-10-12

**Soundness:** 1 poor
**Presentation:** 2 fair
**Contribution:** 2 fair
**Rating:** 1
**Confidence:** 4

**Summary:**

This paper introduces a new approach for addressing both group fairness and individual fairness within groups in graph learning. The authors propose a new definition for individual fairness within groups. The authors introduce a framework named Fairness for Group and Individual (FairGI), which considers both group fairness and individual fairness within groups in the context of graph learning. It employs a similarity matrix of individuals to achieve individual fairness within groups and uses adversarial learning to address group fairness in terms of Equal Opportunity and Statistical Parity. Results from numerous experiments indicate that FairGI outperforms other methods in terms of fairness, while maintaining similar prediction accuracy.

**Strengths:**

1. The authors identify and address a gap in existing research by focusing on both group and individual fairness simultaneously. All the math equations seem correct.
2. The extensive experiments on three datasets with multiple baselines and the ablation study are comprehensive enough.
3. The question this paper asks is important. While there are a lot of works on individual fairness and group fairness, finding a solution to combine them mean something to the community. It is more adaptive than focusing on only individual fairness or only group fairness.

**Weaknesses:**

1. Individual and group fairness are essentially different and probably contradicting objectives, unless the authors discuss why they can be compatible with each other. Even though the authors propose to consider both and try to achieve a better balance by enforcing "individual fairness within groups" instead of "individual fairness for all populations", this conflict could still confuse people and make the motivation less reasonable. The authors only make a statement in the "Challenges of balancing individual fairness and group fairness" paragraph of Section 4.2 while do not make sense to readers why this is reasonable. For example, wouldn't "individual fairness within groups" be essentially still different from "individual fairness" by ignoring individual fairness across groups? Post-hoc experimental analysis is currently not enough to answer this question. There is no explanation or theoretical insights why combining the two conflicting concepts can work well, but they are in fact at the core of supporting the motivation of this problem.
2. Compared to some existing works on fairness such as FairGNN, this work seems quite trivial since people can directly modify the loss objective of FairGNN (https://arxiv.org/pdf/2009.01454.pdf) or DebiAN (https://arxiv.org/pdf/2207.10077.pdf) to incorporate group fairness objectives. Unless authors can such modification to baselines in the comparison experiments and find out that even the modified version cannot compete with the proposed model, I can not convince myself why this work is giving a SOTA solution.
3. The motivation explanation with the toy example in Figure 1 seems strange. Why do group fairness objective select U2 and U3, instead of U3 and U4, since they both get SP=EO=0? All those numbers are only distances, but they do not indicate "score" or "grade". Seems to me U4 is qualified as well, so why not admit U4? Is there any reason that group fairness would tend to select U2 and U3?
4. Moreover, the language used in this paper can potentially stir public outrage: what does “qualified student” mean? I believe the authors do not mean that some students are not “qualified” because they are not "good" in the current system. What if it turns out that something like "a good family" is used as a positive feature in your embedding, since gender is the only considered sensitive factor? How to define “good”? The whole meaning of algorithmic fairness is to eliminate systematic bias and discrimination, and we cannot achieve that if the authors cannot make sense in the core example, regardless of whatever experimental results the paper has shown. It could be the "not even wrong" situation. The way this example puts makes the discussion quite sensitive, if not controversial. Most universities receive a lot of qualified admissions each year, and they never refer to any students as incompetent or unqualified as long as they pass the minimum requirement. At least for this example and for the explanation, I am not convinced and think it can be inappropriate. Please seriously reconsider this example in your paper.

**Questions:**

1. Why would it be possible that after enforcing fairness objectives, "FairGNN" and "Ours" achieve better predictive performance on Acc and AUC than the original GCN and GAT? GCN and GAT are directly designed to minimize the loss and increase Acc/AUC, while fairness objectives are constraints that do not align with the goal of increasing Acc/AUC. I believe even in the original work of "FairGNN", the predictive performance they report is also lower than GCN and GAT baselines. In other works on fairness, I have not personally seen any works reporting such a result. If with good faith that the reported results are correct, the authors must spend more time to analyze this strange phenomenon, which could be fascinating to the community.
Think about it: if we can improve predictive performance by improving fairness, wouldn't it be too perfect to be true? This could be a huge discovery, even way more important than the core contribution of balancing individual/group fairness mentioned in this paper! If it is true across all datasets, this paper should be nominated as the best paper.

2. It would be more interesting to include some theoretical insights and sound examples of why group and individual fairness may not be conflicting or can be optimized concurrently, at least in certain cases. Currently, whether it is sound to optimize them together is not very clear. In this paper, authors also admit this issue but do not give enough analyzes why "individual fairness within groups" can solve this problem. This paper needs a serious reconsideration for submission and a more careful review before submission.

**Details Of Ethics Concerns:**

The way the student admission model is put up does not make sense and can be quite controversial. The authors do not explain how is “qualified” defined, and only uses gender as sensitive attributes. Even though the example is only used to explain why the proposed method is useful, the way it is shown is still not professional, especially if we compare this to some great sociology papers and Nature papers on algorithmic bias. It should not be an excuse to raise a questionable example in a paper, simply because the submission is in the computer science domain.

---

### Official Review · Reviewer_dYzA · 2023-10-30

**Soundness:** 2 fair
**Presentation:** 3 good
**Contribution:** 1 poor
**Rating:** 3
**Confidence:** 4

**Summary:**

The paper introduces a new fairness concept and a framework named FairGI, addressing both fairness types within graph learning. By utilizing a similarity matrix and adversarial learning, FairGI achieves individual fairness within groups and group fairness respectively, showing promising results in fairness metrics and prediction accuracy compared to other leading models.

**Strengths:**

- The paper is clearly written.
- The paper includes thorough experiments to test the algorithm's performance.

**Weaknesses:**

The paper appears to lack substantial original contribution, seemingly combining ideas from two prior works:
- The adversarial training framework for fairness is drawn from "Say No to Discrimination: Learning Fair Graph Neural Networks with Limited Sensitive Attribute Information."
- The computation of the similarity matrix of individuals to ensure individual fairness is inspired by "BeMap: Balanced Message Passing for Fair Graph Neural Network."

Given this synthesis of pre-existing concepts without significant novel insight or advancement, the paper may not meet the high standards of originality and innovation typically expected for acceptance at ICLR.

**Questions:**

I have no question.

---

### Official Review · Reviewer_9gSh · 2023-10-31

**Soundness:** 2 fair
**Presentation:** 2 fair
**Contribution:** 3 good
**Rating:** 5
**Confidence:** 4

**Summary:**

This paper presents a method for simultaneously optimizing group fairness as well as individual fairness within groups for the node classification task. This seems to be the first method that considers both group fairness and individual fairness on a graph. The FaiGI framework designed by the authors has four parts: a GNN classifier, a sensitive attribute classifier, an adversary layer, and an individual fairness module. From the experimental results, the proposed method seems to significantly improve intergroup fairness and individual fairness without sacrificing much accuracy.

**Strengths:**

1. The proposed framework FairGI can optimize the group fairness and individual fairness of GNN simultaneously, and also provide theoretical proof for the convergence of the model;
2. Proposing an evaluation metric for measuring individual fairness within groups;
3. Referring to the experimental results (Table 1), the proposed FaiGI can significantly improve the inter-group fairness and individual fairness without sacrificing too much accuracy;

**Weaknesses:**

1. The design of the model is very similar to FairGNN[1] and the model is not innovative. FairGNN[1] includes 3 parts: a GNN classifier, a sensitive attribute classifier, an adversary layer, so I think FairGI just simply adds individual fairness and EO-related losses to FairGNN[1].
2. Lack of comparison with the latest group fairness methods in the experiment. I suggest the authors to consider comparing with the latest group fairness methods like FairVGNN[3] or EDITS[4].
3. The hyperparameters of the model are excessive, which poses a challenge for tuning. I found a total of 6 fairness-related hyperparameters, and the optimal hyperparameters are different for different datasets (Table 3), which requires a lot of time and computational resources for tuning the parameter when applying different GNN backbones as well as datasets.
4. The experiments lack sensitivity analysis experiments on hyperparameters. Although the experimental results show that FairGI can achieve good results, there are six hyperparameters related to fairness involved in the method (Table 3), but the whole experimental part does not have sensitivity analysis for any of them, which I think is necessary.

Reference

[1] Dai, E., & Wang, S. (2021, March). Say no to the discrimination: Learning fair graph neural networks with limited sensitive attribute information. In Proceedings of the 14th ACM International Conference on Web Search and Data Mining (pp. 680-688).

[2] Agarwal, Chirag, Himabindu Lakkaraju, and Marinka Zitnik. "Towards a unified framework for fair and stable graph representation learning." Uncertainty in Artificial Intelligence. PMLR, 2021.

[3] Wang, Y., Zhao, Y., Dong, Y., Chen, H., Li, J., & Derr, T. (2022). Improving Fairness in Graph Neural Networks via Mitigating Sensitive Attribute Leakage. In Proceedings of the 28th ACM SIGKDD Conference on Knowledge Discovery and Data Mining (pp. 1938–1948).

[4] Dong, Y., Liu, N., Jalaian, B., & Li, J. (2022, April). Edits: Modeling and mitigating data bias for graph neural networks. In Proceedings of the ACM Web Conference 2022 (pp. 1259-1269).

**Questions:**

1. Add latest baseline on group fairness.
2. Add experiments on hyperparameter sensitivity analysis.
3. The definition of symbols in the problem definition section (3.2.2) regarding the grouping of different sensitivity attributes is not uniform.
4. "L is the Laplacian matrix of M" in Definition 4, but I didn't find the symbol L in Publication 7.
5. There is no introduction of "IF" in the introduction of evaluation indexes in the experimental part.

---

### Official Review · Reviewer_n2zH · 2023-10-31

**Soundness:** 2 fair
**Presentation:** 3 good
**Contribution:** 1 poor
**Rating:** 3
**Confidence:** 4

**Summary:**

This paper proposes a novel framework called Fairness for Group and Individual (FairGI) that addresses both group and individual fairness within groups in the context of graph learning. The approach employs the similarity matrix of individuals to achieve individual fairness within groups, while leveraging adversarial learning to address group fairness in terms of both Equal Opportunity and Statistical Parity. The primary contributions of this paper are: (1) introducing a novel problem concerning the achievement of both group fairness and individual fairness within groups in graph learning, (2) proposing a new metric to measure individual fairness within groups for graphs, (3) proposing an innovative framework FairGI to ensure group fairness and individual fairness within groups in graph learning and maintaining comparable model prediction performance, and (4) demonstrating the effectiveness of the framework in eliminating both group and individual fairness and maintaining comparable prediction performance through comprehensive experiments on various real-world datasets.

**Strengths:**

1. The authors propose a group fairness and in-group individual fairness problem.
2. The written of the paper is clear.
3. The authors design clear experiments to demonstrate the advantage of the proposed method.

**Weaknesses:**

1. The contribution appears incremental, primarily integrating individual fairness loss and group fairness loss from existing studies without significant innovation.
2. The definition and input of the $L_p$ function in Equation 9 are ambiguous, as both $Z$ and $f$ are presented as inputs. Clarification is needed regarding the role and relationship of these variables within the function.
3. The results are limited to the Pokec-n dataset. Given that Pokec-z is a similarly prevalent dataset, its omission is notable. The inclusion of Pokec-z results could enhance the robustness and generalizability of the findings.
4. The trade-offs between utility performance, group fairness, and individual fairness are not adequately explored. A Pareto frontier representation could provide valuable insights into the interplay between these dimensions.
5. There are typographical errors in Section 3.2.2; specifically, the second line features two instances of $V_{p_2}$ that seem out of place and likely constitute errors.

**Questions:**

In weakness

---

### Meta-Review · Area_Chair_wGMK · 2023-12-05

**Metareview:**

This paper proposes a novel framework called Fairness for Group and Individual (FairGI) that addresses both group and individual fairness within groups in the context of graph learning. The approach employs the similarity matrix of individuals to achieve individual fairness within groups, while leveraging adversarial learning to address group fairness in terms of both Equal Opportunity and Statistical Parity. The primary contributions of this paper are: (1) introducing a novel problem concerning the achievement of both group fairness and individual fairness within groups in graph learning, (2) proposing a new metric to measure individual fairness within groups for graphs, (3) proposing an innovative framework FairGI to ensure group fairness and individual fairness within groups in graph learning and maintaining comparable model prediction performance, and (4) demonstrating the effectiveness of the framework in eliminating both group and individual fairness and maintaining comparable prediction performance through comprehensive experiments on various real-world datasets. Specifically, the strength of this paper includes several aspects. 1) The authors propose a group fairness and in-group individual fairness problem. 2) The written of the paper is clear. 3) The authors design clear experiments to demonstrate the advantage of the proposed method.

However, there are several points to be further improved. For example, the design of the model is very similar to FairGNN and the model is not innovative. FairGNN includes 3 parts: a GNN classifier, a sensitive attribute classifier, an adversary layer, so FairGI just simply adds individual fairness and EO-related losses to FairGNN. Comparison with the latest group fairness methods should be added in the experiment. The authors should consider comparing with the latest group fairness methods like FairVGNN or EDITS. The hyperparameters of the model are excessive, which poses a challenge for tuning. A total of 6 fairness-related hyperparameters, and the optimal hyperparameters are different for different datasets (Table 3), which requires a lot of time and computational resources for tuning the parameter when applying different GNN backbones as well as datasets. The experiments lack sensitivity analysis experiments on hyperparameters. Although the experimental results show that FairGI can achieve good results, there are six hyperparameters related to fairness involved in the method (Table 3), but the whole experimental part does not have sensitivity analysis for any of them. Therefore, this paper cannot be accepted at ICLR this time, but the enhanced version is highly encouraged to submit other top-tier venues.

**Justification For Why Not Higher Score:**

However, there are several points to be further improved. For example, the design of the model is very similar to FairGNN and the model is not innovative. FairGNN includes 3 parts: a GNN classifier, a sensitive attribute classifier, an adversary layer, so FairGI just simply adds individual fairness and EO-related losses to FairGNN. Comparison with the latest group fairness methods should be added in the experiment. The authors should consider comparing with the latest group fairness methods like FairVGNN or EDITS. The hyperparameters of the model are excessive, which poses a challenge for tuning. A total of 6 fairness-related hyperparameters, and the optimal hyperparameters are different for different datasets (Table 3), which requires a lot of time and computational resources for tuning the parameter when applying different GNN backbones as well as datasets. The experiments lack sensitivity analysis experiments on hyperparameters. Although the experimental results show that FairGI can achieve good results, there are six hyperparameters related to fairness involved in the method (Table 3), but the whole experimental part does not have sensitivity analysis for any of them. Therefore, this paper cannot be accepted at ICLR this time, but the enhanced version is highly encouraged to submit other top-tier venues.

**Justification For Why Not Lower Score:**

However, there are several points to be further improved. For example, the design of the model is very similar to FairGNN and the model is not innovative. FairGNN includes 3 parts: a GNN classifier, a sensitive attribute classifier, an adversary layer, so FairGI just simply adds individual fairness and EO-related losses to FairGNN. Comparison with the latest group fairness methods should be added in the experiment. The authors should consider comparing with the latest group fairness methods like FairVGNN or EDITS. The hyperparameters of the model are excessive, which poses a challenge for tuning. A total of 6 fairness-related hyperparameters, and the optimal hyperparameters are different for different datasets (Table 3), which requires a lot of time and computational resources for tuning the parameter when applying different GNN backbones as well as datasets. The experiments lack sensitivity analysis experiments on hyperparameters. Although the experimental results show that FairGI can achieve good results, there are six hyperparameters related to fairness involved in the method (Table 3), but the whole experimental part does not have sensitivity analysis for any of them. Therefore, this paper cannot be accepted at ICLR this time, but the enhanced version is highly encouraged to submit other top-tier venues.

---

### Decision · Program_Chairs · 2024-01-16

Reject